# Analysis of the evolution of resistance to multiple antibiotics enables prediction of the *Escherichia coli* phenotype-based fitness landscape

**Junichiro Iwasawa**[1], **Tomoya Maeda**[2], **Atsushi Shibai**[3], **Hazuki Kotani**[3], **Masako Kawada**[3], **Chikara Furusawa**[1,3,4] *

1 Department of Physics, Graduate School of Science, University of Tokyo, Tokyo, Japan, 2 Graduate School of Agriculture Research, Faculty of Agriculture, Hokkaido University, Sapporo, Japan, 3 Center for Biosystems Dynamics Research, RIKEN, Suita, Japan, 4 Universal Biology Institute, Graduate School of Science, University of Tokyo, Tokyo, Japan

* chikara.furusawa@riken.jp

**Data Availability Statement:** Data and code for the reconstruction of all figures including the phenotype-fitness landscape are available via

## Abstract

The fitness landscape represents the complex relationship between genotype or phenotype and fitness under a given environment, the structure of which allows the explanation and prediction of evolutionary trajectories. Although previous studies have constructed fitness landscapes by comprehensively studying the mutations in specific genes, the high dimensionality of genotypic changes prevents us from developing a fitness landscape capable of predicting evolution for the whole cell. Herein, we address this problem by inferring the phenotype-based fitness landscape for antibiotic resistance evolution by quantifying the multidimensional phenotypic changes, i.e., time-series data of resistance for eight different drugs. We show that different peaks of the landscape correspond to different drug resistance mechanisms, thus supporting the validity of the inferred phenotype-fitness landscape. We further discuss how inferred phenotype-fitness landscapes could contribute to the prediction and control of evolution. This approach bridges the gap between phenotypic/genotypic changes and fitness while contributing to a better understanding of drug resistance evolution.

## Introduction

Laboratory evolution experiments, high-throughput sequencing, and phenotyping combined with data analysis have heralded a new age for evolutionary biology [1–3]. Massively parallel laboratory evolution experiments have revealed an unprecedented amount of information on evolution, including epistatic interactions in genotypic space [4–7], genotype–phenotype mapping [8–14], and the existence of repeatable features in evolutionary processes [8,15–17]. Furthermore, quantitative and theoretical modelling have enabled us to build fitness models based on genotypic and phenotypic data [2,18–20], enabling us to explore the origins of the constraints underlying evolution [21–25]. Despite numerous attempts to describe evolutionary dynamics, predicting and controlling evolution remains difficult.

https://github.com/jiwasawa/resistance-landscape.
The reseIC_quencing analysis data have been
deposited at the DDBJ Sequence Read Archive
(https://ddbj.nig.ac.jp/DRASearch/) under
accession number DRA014660.

**Funding:** This study was supported in part by
JSPS KAKENHI (17H06389 and 19H05626 to C.F.;
https://www.jsps.go.jp/english/index.html), JST
ERATO (JPMJER1902 to C.F.; https://www.jst.go.
jp/erato/en/index.html). J.I. was supported by a
Grant-in-Aid for the Japan Society for Promotion of
Science Fellows (JP18J21942). The funders had
no role in study design, data collection and
analysis, decision to publish, or preparation of the
manuscript.

**Competing interests:** The authors have declared
that no competing interests exist.

**Abbreviations:** IS, insertion sequence; KM,
kanamycin; MMC, mitomycin C; MSE, mean
squared error; NFLX, norfloxacin; NQO, 4-
nitroquinoline 1-oxide; OD, optical density; PCA,
principal component analysis; PLM, phleomycin;
PMF, proton motive force; SDC, sodium
dichromate dihydrate; SS, sodium salicylate; TET,
tetracycline.

Fitness (or adaptive) landscapes are frequently invoked in experimental and theoretical studies as they offer a basis for the predictability and convergence of evolution [26,27]. Thus, constructing empirical fitness landscapes based on experimental data may be a suitable approach for predicting evolution. Most previous studies on empirical fitness landscapes were based on fitness assays of large mutant libraries for a gene of interest, which revealed broad interactions between mutations and the nonlinearity of the underlying landscapes [4,6,7,28,29]. Despite our accumulating knowledge on how specific mutations affect fitness, we are still far from constructing a comprehensive fitness landscape capable of predicting the evolutionary process of an entire cell. This difficulty is because of the high dimensionality of genotypic space, which makes it difficult to acquire sufficient data for prediction [2,30]. Parallel laboratory evolution experiments have shown that different single nucleotide changes can underlie similar phenotypic changes, implying the existence of multiple paths in the genotypic space to reach a fitted phenotype [14,31–33]. However, phenotypes often exhibit repeatable features in laboratory evolution, implying the low dimensionality of phenotypic space for evolution [2,8,15–18]. Previous studies thus suggest that the fitness landscape may be more traceable when using phenotypes rather than high-dimensional genotypes as its basis.

In this study, we aimed to construct an empirical fitness landscape for the evolution of antibiotic resistance in *Escherichia coli* using phenotypes as its basis. Our basic strategy was to densely observe multiple phenotypes and their corresponding fitness (i.e., drug resistance) along different trajectories of evolution (Fig 1A and 1B). The observation of trajectories under different antibiotics, starting from different locations in phenotypic space, could allow us to sample sufficient phenotypes to help build the phenotype-fitness landscape. The critical feature of our work is that we used eight different antibiotic resistance values ($IC_{50}$) as probes for fitness, and their principal components as phenotypes (Fig 1B). As previous studies have suggested, the antibiotic resistance space corresponds to a subspace of the gene expression space, making it a good candidate for probing evolutionary dynamics [14,18]. In addition, antibiotic resistance values can be measured in a high-throughput manner [34], enabling the efficient sampling of phenotypes during the course of laboratory evolution.

## Results

### Laboratory evolution with multi-antibiotic resistance measurements

We used seven starting strains for laboratory evolution to observe a wide range of phenotypic changes associated with the evolution of antibiotic resistance (Fig 1A). Six out of the seven starting strains had drug-resistant phenotypes, obtained by laboratory evolution over approximately 300 generations [14]. For example, KME1 and KME5 strains evolved from independent culture series under the selection of the antibiotic kanamycin (KM). In our previous study, we confirmed that these strains showed increased $IC_{50}$ values for KM and exhibited cross-resistance and collateral sensitivity to various drugs [14]. In addition to KM-, norfloxacin (NFLX)-, and tetracycline (TET)-resistant strains, we used the parent strain of a previous study, *E. coli* MDS42, as one of the starting strains.

We performed laboratory evolution from these seven starting strains under the selection of three antibiotics: TET, KM, and NFLX (Fig 1A). The daily passage occurred in 384-well microplates for every approximately 10 generations using an automated culture system (see Materials and methods). We also quantified the $IC_{50}$ for multiple drugs during every passage to capture phenotypic changes during laboratory evolution. In addition to TET, KM, and NFLX, we adopted the following five drugs (other than antibiotics): sodium salicylate (SS), phleomycin (PLM), 4-nitroquinoline 1-oxide (NQO), sodium dichromate dihydrate (SDC), and mitomycin C (MMC), to characterize the phenotypic changes during adaptive evolution. We selected these drugs to optimize the expressiveness of phenotypic changes during drug

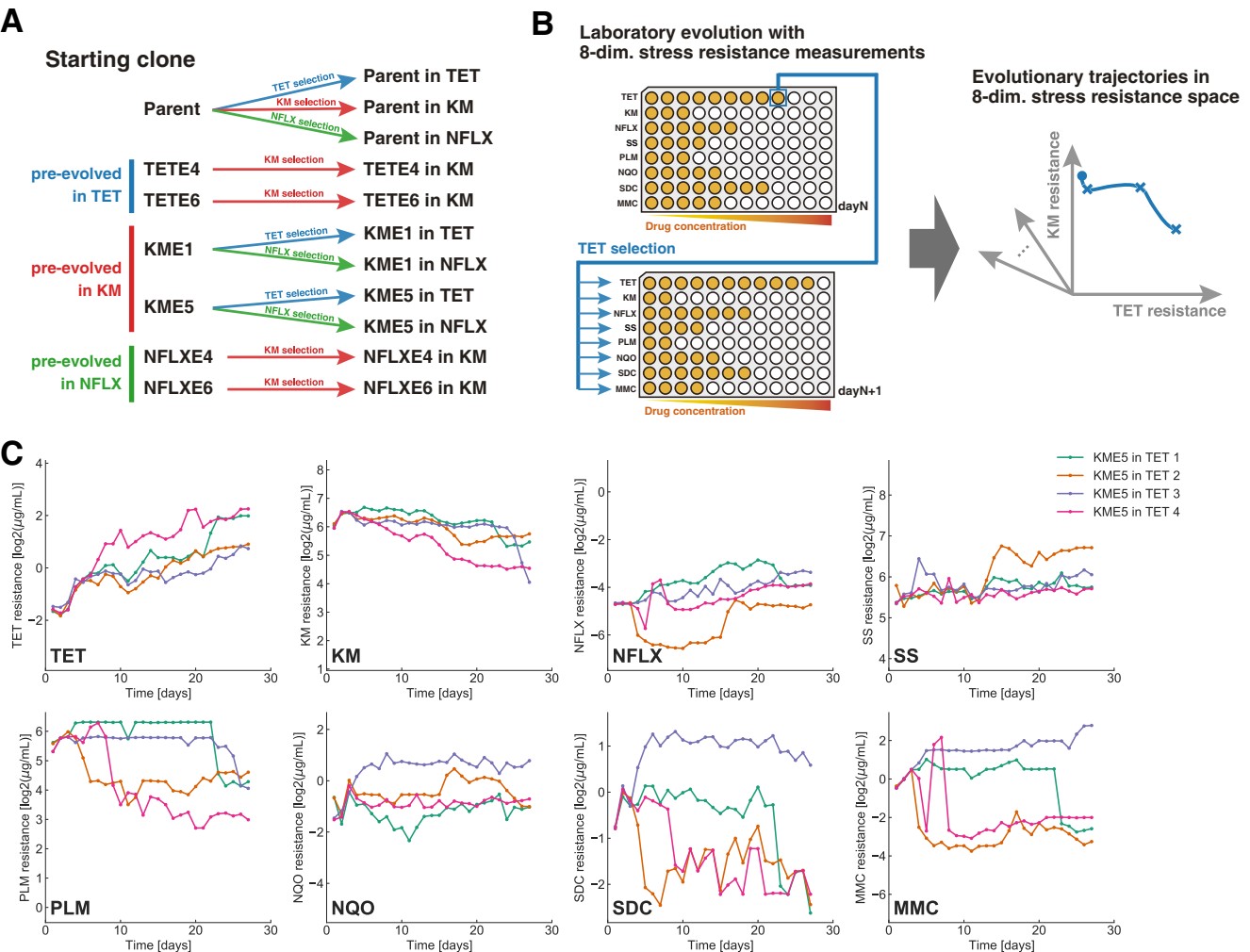

**Fig 1. Laboratory evolution of *Escherichia coli* to probe evolutionary trajectories in a multidimensional resistance space.** (**A**) The experimental conditions used in the current study. We prepared seven starting strains, i.e., the parent *E. coli* strain, evolved under TET (TETE4, TETE6), evolved under KM (KME1, KME5), and evolved under NFLX (NFLXE4, NFLXE6). Starting from these strains, we performed laboratory evolution under three drugs (TET, KM, and NFLX). (**B**) Schematic figure describing the protocol of the laboratory evolution experiment with simultaneous $IC_{50}$ measurements along the evolutionary trajectories in the present study. For every passage, $IC_{50}$ was quantified for eight drugs, and cells were transferred from the well with the highest selection-drug concentration where the cell concentration exceeded a threshold. (**C**) The time series of $IC_{50}$ values during the evolution of the KME5 strain under TET, for example. Data of four independent culture lines are overlaid. The resistance values for TET, KM, NFLX, SS, PLM, NQO, SDC, and MMC are shown. All time-series data in the present study are presented in S1 Fig. The data underlying this figure can be found in S1 Data. KM, kanamycin; MMC, mitomycin C; NFLX, norfloxacin; NQO, 4-nitroquinoline 1-oxide; PLM, phleomycin; SDC, sodium dichromate dihydrate; SS, sodium salicylate; TET, tetracycline.

resistance evolution, based on the transcriptome and $IC_{50}$-values of drug-resistant strains obtained in a previous study [14] (see Materials and methods for details).

In Fig 1C, we show the time series of $IC_{50}$ for the eight drugs investigated, starting from the KM-resistant strain (KME5) under TET selection; for example, the data of four independent culture lines are overlaid. We performed laboratory evolution for 11 combinations of starting strains and selection drugs (Fig 1A), resulting in 44 trajectories in the eight-dimensional drug resistance space (see S1 Fig for all time-series data). As shown in Fig 1C, selection by TET significantly changed the resistance to other drugs by cross-resistance and collateral sensitivity. For example, the strains presented decreased $IC_{50}$ for KM and PLM when the TET resistance increased, indicating a collateral sensitivity relationship between these drugs.

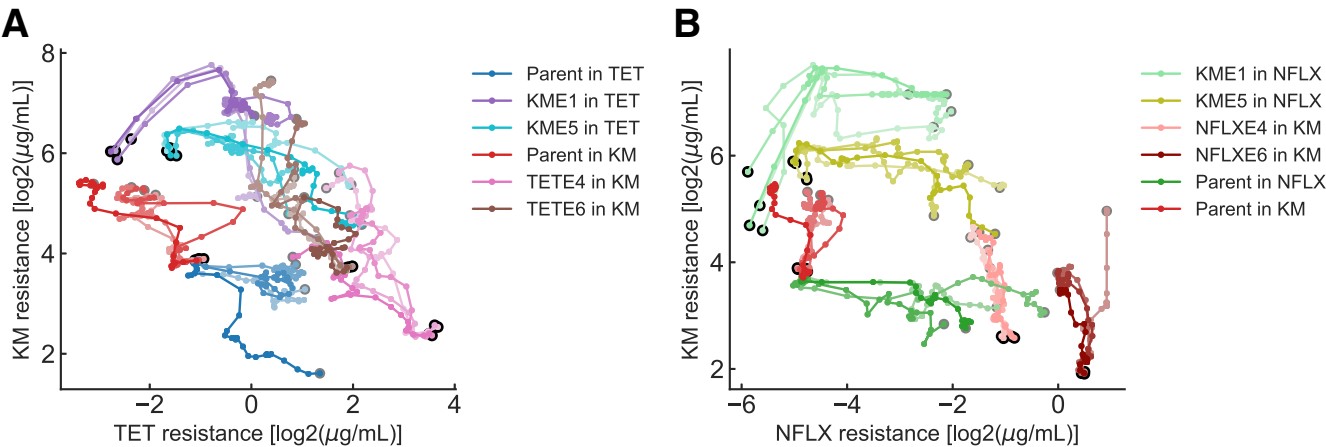

**Fig 2. Evolutionary trajectories in two-dimensional drug resistance space.** (**A**) Evolutionary trajectories in TET–KM resistance space. The horizontal and vertical axes show the log-transformed $IC_{50}$ values for TET and KM, respectively. The black and gray circles represent the states of starting strains and the last day of evolution, respectively. (**B**) Evolutionary trajectories in NFLX–KM resistance space. The data underlying this figure can be found in S1 Data. KM, kanamycin; NFLX, norfloxacin; TET, tetracycline.

It should be noted that the four independent culture series did not always show similar trajectories for resistance acquisition and loss. For example, the time development of $IC_{50}$ for SDC and MMC in Fig 1C exhibited heterogeneous patterns, wherein one strain acquired SDC and MMC resistance, whereas the other three acquired sensitivity to these antibiotics. The genetic background underlying this heterogeneity in the development of resistance is discussed later.

The trajectories in the resistance space revealed a relationship between resistance acquisition and loss for multiple drugs. For example, the two-dimensional TET-KM resistance space showed a trade-off relationship between the KM and TET resistances (Fig 2A). A trade-off relationship between TET and KM has been reported in previous studies [11,14,18]. In contrast to the negatively correlated trajectories in the TET-KM space, resistance to some antibiotic pairs appeared to be acquired independently. In Fig 2B, we show the evolutionary trajectories in the NFLX-KM antibiotic resistance space. Here, the evolutionary trajectories that started from NFLX evolution seemed to acquire resistance to KM with little loss of resistance to NFLX and vice versa. The observation that resistance acquisition for NFLX and KM could occur independently suggests that the resistance acquisition mechanisms for the two antibiotics are modular [24].

## Observing trajectories in the PCA space

We performed principal component analysis (PCA) of the resistance profiles for the 44 trajectories to investigate the evolutionary trajectories in the eight-dimensional resistance space. Because the dynamic ranges of $IC_{50}$ varied among the eight antibiotics, we normalized them and set the mean and standard deviation of the $IC_{50}$ variation among the 44 trajectories over the 27 days of evolution to (mean, standard deviation) = (0,1) before applying PCA. In Fig 3A and 3B, examples of evolutionary trajectories are highlighted in two-dimensional PCA space (explained variance ratio PC1:41%, PC2:23%). While most of the strains took similar trajectories, indicating convergence in the phenotypic space, several conditions (such as strains under TET evolution starting from KME1 and KME5) exhibited different evolutionary paths (Fig 3A). This divergence for TET evolution and the underlying genetic backgrounds will be discussed later. To interpret the phenotypes in PCA space, we plotted the $IC_{50}$ values for each

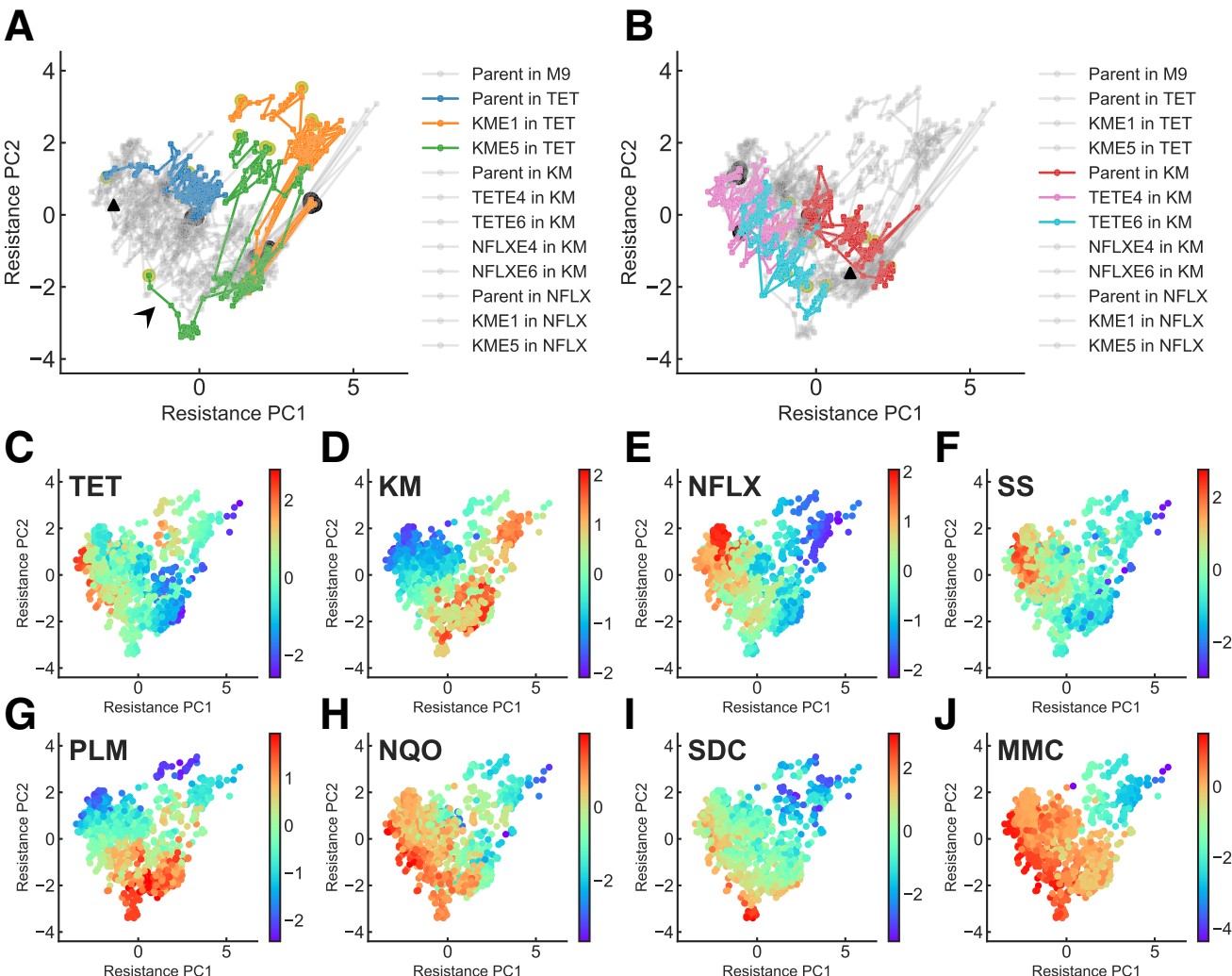

**Fig 3. Evolutionary trajectories in two-dimensional PCA space.** A total of 44 trajectories (11 conditions with four replicates each) in an eight-dimensional resistance space were projected into two-dimensional space using PCA. The variance explained by the first two components was 41% and 23%, respectively. (**A**) Evolutionary trajectories under TET selection. Trajectories starting from the parent and KM-evolved strains are highlighted by colors, while the gray lines show the trajectories in other conditions. The black and yellow circles represent the state of starting strains and the strains on the last day, respectively. (**B**) Evolutionary trajectories under KM selection. Trajectories starting from the parent and TET-evolved strains under KM selection are highlighted in color. The black triangles in (**A**) and (**B**) show the peaks of the inferred phenotype-fitness landscape for TET and KM, respectively, and will be explained later in Fig 4. (**C–J**) IC$_{50}$ values for each drug mapped on the two-dimensional PCA space. The colors represent the log-transformed IC$_{50}$ values (log$_2$ μg/mL). The data underlying this figure can be found in S1 Data. KM, kanamycin; PCA, principal component analysis; TET, tetracycline.

antibiotic (Fig 3C–3J). These plots allowed us to visualize phenotypes with high fitness for the corresponding drugs. For example, the phenotypes with high KM resistance were located on the lower-right side of the PCA space, indicating that PC1 is positively correlated with KM resistance (Fig 3D). This tendency is also consistent with the evolutionary trajectories under KM selection (Fig 3B). The results indicate that the phenotypes that were highly resistant to each drug occupied different regions of PCA space.

## Inference of the phenotype-fitness landscapes

Based on the analysis shown in Fig 3, we inferred the phenotype-fitness landscape of drug resistance on the two-dimensional PCA plane by smoothing the IC$_{50}$ data via a Gaussian

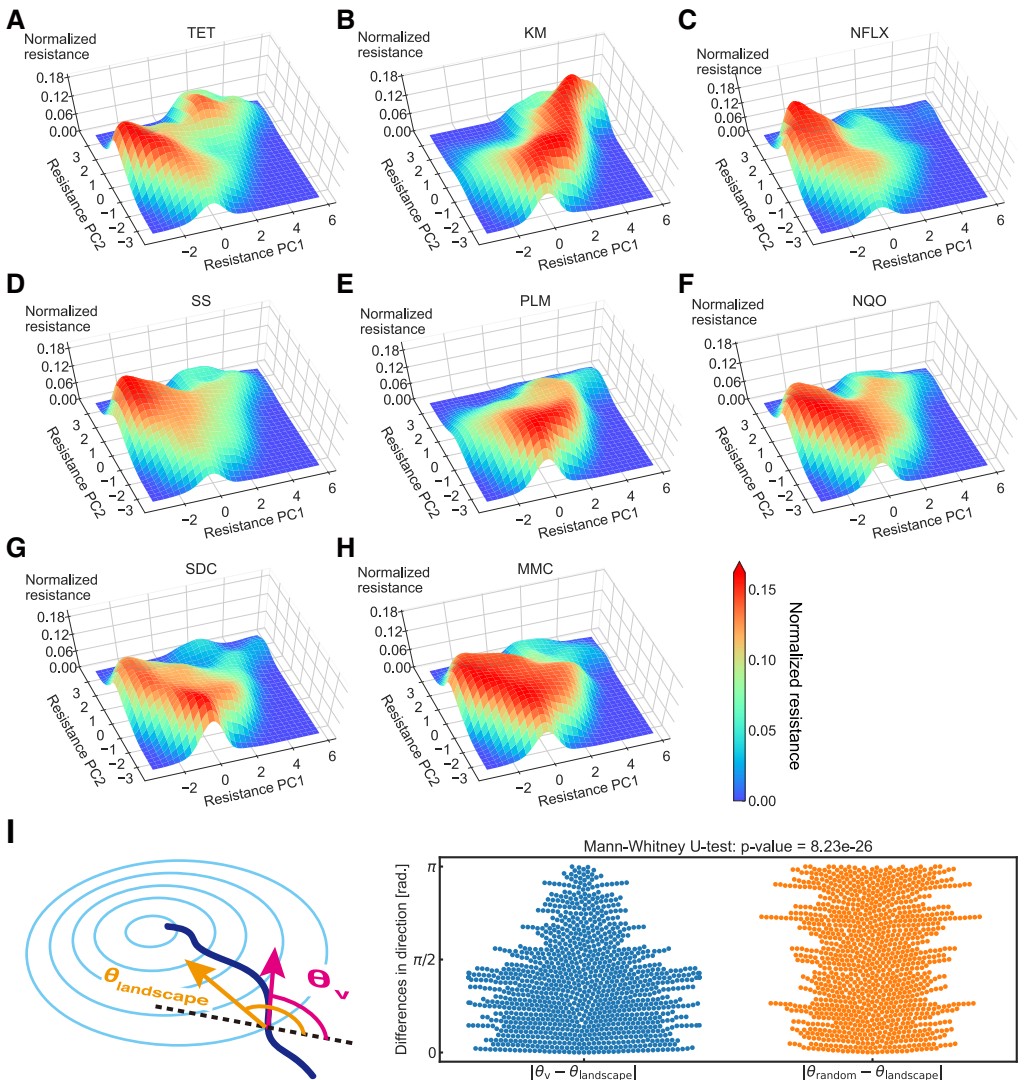

**Fig 4. Estimated phenotype-fitness landscape. (A–H)** The phenotype-fitness landscape obtained using Gaussian smoothing for eight drugs is presented. The resistance levels were normalized to a maximum value of one (see Materials and methods). (**I**) Differences in the evolutionary directions between the simulated gradient from the phenotype-fitness landscape ($\theta_{\text{landscape}}$) and experimental observation ($\theta_v$). For comparison, the difference with a random direction ($\theta_{\text{random}}$) sampled from a uniform distribution $[0,2\pi]$ is also plotted. The data underlying this figure can be found in S1 Data. KM, kanamycin; MMC, mitomycin C; NFLX, norfloxacin; NQO, 4-nitroquinoline 1-oxide; PLM, phleomycin; SDC, sodium dichromate dihydrate; SS, sodium salicylate; TET, tetracycline.

kernel (see Materials and methods for details). Fig 4 shows the inferred phenotype-fitness landscapes for the eight antibiotics used in this study. The inferred landscapes have a small number of peaks for each landscape, suggesting low ruggedness [4,7]. Note that the high noise frequency in the original $IC_{50}$ time series, compared with the mutation acquisition frequency, implies that the noise stems from measurement errors rather than from genotypic ruggedness.

We first evaluated whether the landscape could represent the observed evolutionary trajectories by analyzing the relationship between the gradients in the landscape and the direction of phenotypic changes. As the raw evolutionary trajectories were subject to experimental noise, we applied a moving average of six time points to the trajectories using a triangular window.

The direction provided by the landscape $\theta_{\text{landscape}}(\boldsymbol{x})$ at location $\boldsymbol{x}$ is calculated as follows:

$$\theta_{\text{landscape}}(\boldsymbol{x}) = \tan^{-1} \frac{\partial F(\boldsymbol{x})/\partial x_{\text{PC2}}}{\partial F(\boldsymbol{x})/\partial x_{\text{PC1}}}, \tag{1}$$

where $F(\boldsymbol{x})$ represents the altitude of the phenotype-fitness landscape and $\partial/\partial x_{\text{PC1,2}}$ is the gradient along the PC1,2 axis (Fig 4I). $\theta_{v}(i)$, the evolutionary direction in the experimental observations, was calculated from the coordinates of the $i$th time point in the smoothed trajectories $x_{\text{PC1,2}}(i)$ using

$$\theta_{v}(i) = \tan^{-1} \frac{x_{\text{PC2}}(i+1) - x_{\text{PC2}}(i)}{x_{\text{PC1}}(i+1) - x_{\text{PC1}}(i)}. \tag{2}$$

In Fig 4I, we plotted the distribution of $|\theta_{v}-\theta_{\text{landscape}}|$ where $\theta_{\text{landscape}}$ was calculated for each corresponding point along the observed trajectories. For comparison, we also plotted $|\theta_{v}-\theta_{\text{random}}|$, where $\theta_{\text{random}}$ was sampled from a uniform distribution $[0,2\pi]$. As shown in Fig 4I, the directions of evolution provided by the inferred landscape $\theta_{\text{landscape}}$ had a significant correspondence with the observed trajectories $\theta_{v}$ ($p = 9.23\times10^{-26}$, one-sided Mann–Whitney U test), suggesting that the inferred landscapes could reproduce the observed evolutionary trajectories. Note that given the fact that each step along the trajectories was defined by time (day) and not by a step in the genotypic space, there were moments where the trajectories stayed at a certain point and where $\theta_{v}$ deviated from the inferred landscape $\theta_{\text{landscape}}$. In addition, the inferred phenotype-fitness landscape was calculated based only on the first two principal components, and the remaining 36% of the variance was not considered. Although we compared the inferred landscape with the trajectories used for the inference process, the two limitations raised above suggest that the correspondence between $\theta_{v}$ and $\theta_{\text{landscape}}$ was not trivial.

The predictability of the inferred landscapes was evaluated by applying train/test splitting to the whole dataset (S2 Fig). Here, the training dataset was obtained by randomly excluding one out of the four replicates for each experimental condition, resulting in a total of 33 lines, and the excluded 11 lines were used for the test dataset. The phenotype-fitness landscape was inferred using the training data (S2A–S2C Fig) and was compared with the evolutionary directions ($\theta_{\text{landscape,train}}$), with experimental observations based on the test data ($\theta_{v,\text{test}}$) (S2D Fig). The landscape gradients and experimental observations showed significant agreement. Note that this verification for predictability is only possible for the landscape of TET, KM, and NFLX since the laboratory evolution experiments were performed only under these three drugs. However, the landscape properties of the other five drugs might also have similar predictive power due to the tight interconnectedness of drug resistance acquisition relations, which is known as cross-resistance and collateral sensitivity [11,12,14,18]. Overall, the results suggest a correspondence between the landscape-based properties and the experimental observations, indicating the validity and predictive power of the inferred landscapes.

## Genotypes underlying two peaks in TET fitness landscape

One interesting finding in the previous section is that there were two distinct peaks in the TET fitness landscape (Fig 4A). This structure originated from differences in evolutionary trajectories, starting from different initial strains. As shown in Fig 3A, most of the trajectories that started from KM evolved strains evolved toward the upper minor peak to increase TET resistance. In contrast, the phenotypes of other TET-resistant strains (e.g., TETE4 and TETE6 starting strains; see Fig 3B) were close to the major peak. This result demonstrates the historical dependence of the evolution of TET resistance.

**Table 1. Mutations identified in KM evolved strains under TET selection.**

| Strain | Position | Mutation | Annotation | Gene |
|---|---|---|---|---|
| KME1 in TET 1 | 368,605 | $(T)_{5\to6}$ | coding (1,975/2,355 nt) | *lon* → |
| KME1 in TET 2 | 367,435 | Δ 4 bp | coding (805,808/2,355 nt) | *lon* → |
| | 394,075 | C→T | A191V (GCC→GTC) | *acrR* → |
| | 3,348,058 | G→A | S338F (TCT→TTT) | *atpA* ← |
| KME1 in TET 3 | 368,605 | $(T)_{5\to6}$ | coding (1,975/2,355 nt) | *lon* → |
| | 368,605 | A→G | intergenic (186/26) | *marC* ←/→ *marR* |
| KME1 in TET 4 | 368,810 | (GCGCA)1→2 | coding (2,180/2,355 nt) | *lon* → |
| | 1,096,089 | T→G | F140L (TTT→TTG) | *rssB* → |
| | 1,325,928 | A→G | intergenic (193/19) | *marC* ←/→ *marR* |
| | 3,600,861 | A→C | noncoding (12/120 nt) | *rrfB* → |
| | 3,716,065 | G→T | R174S (CGC→AGC) | *acs* ← |
| KME5 in TET 1 | 368,820 | Δ 1 bp | coding (2,190/2,355 nt) | *lon* → |
| | 1,095,758 | T→A | L30* (TTG→TAG) | *rssB* → |
| | 2,986,848 | T→C | E316G (GAG→GGG) | *envZ* ← |
| | 3,707,114 | G→A | G145S (GGC→AGC) | *soxR* → |
| KME5 in TET 2 | 368,633 | (CCACGT)2→3 | coding (2,003/2,355 nt) | *lon* → |
| | 1,390,792 | T→A | L23* (TTG→TAG) | *rsxD* → |
| KME5 in TET 3 | 393,230 | C→T | V45I (GTC→ATC) | *acrA* ← |
| | 2,908,063 | T→G | E273D (GAA→GAC) | *rpoA* ← |
| | 3,350,544 | A→T | W241R (TGG→AGG) | *atpB* ← |
| KME5 in TET 4 | 367,701 | Δ 1 bp | coding (1,071/2,355 nt) | *lon* → |
| | 1,325,907 | Δ 35 bp | intergenic (172/6) | *marC* ←/→ *marR* |
| | 3,124,036 | Δ 1 bp | coding (451/1,950 nt) | *yhjK* ← |
| | 3,345,579 | G→A | P337L (CCG→CTG) | *atpD* ← |
| | 3,969,252 | G→T | A201E (GCG→GAG) | *rob* ← |

To clarify the genetic mechanism of this historical dependence, we resequenced eight TET-evolved strains starting from KM-evolved strains (four from KME1 and four from KME5) to identify mutation fixation during evolution under TET selection. As shown in Table 1, 7/8 of the resequenced strains had null mutations in the coding region of *lon*, which encodes Lon protease. These seven strains coincided with the strains that evolved toward the minor peak in the landscape, suggesting that the minor peak observed in Fig 4A corresponds to resistance acquisition caused by a *lon* mutation. It has been previously shown that the *lon*⁻ mutant shows a low level of multidrug resistance, considering TET, ampicillin, chloramphenicol, and erythromycin (see Table 1 in [35] and [36,37]). We also confirmed the effect of the *lon* mutation on the parent MDS42 strain, showing that the *lon*⁻ mutant exhibited a 2-fold increase in $IC_{50}$ for TET [14]. This study also showed that the *lon*⁻ mutation caused sensitivity to MMC, which is consistent with the fact that the minor peak was located in the MMC-sensitive region (Fig 3J).

Interestingly, while *lon* mutations were identified in most of the TET-evolved strains starting from the KM-resistant phenotype, these mutations were not observed under TET selection when it started from the MDS42 parent strain [14]. In the latter case, mutations in the *acrR* regulator, which can activate the expression of the *acrAB* efflux pump genes, leading to TET resistance, were commonly identified. Here, the increase in $IC_{50}$ owing to the *acrR*⁻ mutation was significantly larger than that due to the null mutation of *lon* [14]. This raises the following question: What makes the *lon* mutation special for TET resistance when starting from a KM-evolved strain?

In *E. coli*, the uptake of aminoglycosides as KM is positively correlated with the proton motive force (PMF) [38,39]. Indeed, the KM-resistant phenotype in KME1 and KME5 strains was achieved by decreasing PMF to suppress the uptake of KM by null mutations of *cyo* genes whose products are involved in the electron transfer system (the identified mutations in KME1 and KME5 are presented in Supplementary Data 3 in [14]). Simultaneously, these *cyo* mutations result in hypersensitivity to several antibiotics, including TET [14]. This hypersensitivity was caused by the decreased activity of the AcrAB efflux pump, as this pump is a proton antiporter, and its activity is positively correlated with PMF [40,41]. The trade-off between KM and TET resistance is based on these PMF-dependent changes in the efflux and uptake activities of drugs [11,42].

In the present study, the KM evolved strains with *cyo* mutations did not acquire *acrR* mutations under TET selection, probably because of the decreased fitness gain of *acrR* mutations after the fixation of the *cyo* mutations. Specifically, increasing the activity of the AcrAB efflux pump by the *acrR* mutation is difficult under the decreased activity of the electron transfer system by *cyo* mutations. As an alternative strategy, in KME1- and KME5-evolved strains, *lon* mutations increased TET resistance. Although the detailed mechanism by which *lon* mutations contribute to TET resistance is unclear, one possible mechanism can be related to the stabilization of the transcriptional activator MarA, which is the substrate of the Lon protease [36,43], and subsequent activation of the *marRAB* operon. It was recently shown that the activation of the *marRAB* operon can cause antibiotic resistance through the acidification of the cytoplasm [44]. This previous study also showed that acidification-based drug resistance was independent of changes in PMF. These results suggest that this acidification-related resistance phenotype is a possible strategy for KME1- and KME5-evolved strains to achieve TET resistance under decreased PMF through *cyo* mutations.

It should be noted that one out of eight TET-evolved strains starting from KM-evolved strains had no mutation in *lon*. The evolutionary trajectory toward this strain exhibited a different direction of phenotypic change (denoted by the arrowhead in Fig 3A) compared with the other seven trajectories. We confirmed that this strain had acquired mutations in *acrA*, *rpoA*, and *atpB*. The identified V45I mutation in *acrA* corresponded to the MP domain of AcrA, which transmits conformational changes in AcrB to TolC [45]. Given these results, we speculate that this strain acquired resistance to TET by improving the activity of the AcrAB-TolC efflux pump through an *acrA* mutation and not through the *lon* mutation.

## Utilizing landscapes for controlling evolution

We inferred the fitness landscapes of phenotypic changes (Fig 4) and determined their consistency with the observed evolutionary trajectories. The next step was thus to utilize the landscape to control evolution. The basis of the inferred phenotype-fitness landscape was given by the two principal components calculated from the IC$_{50}$ values of the eight drugs. This means that if we monitor the resistance of the eight drugs during the course of evolution, we may be able to compute their location in the phenotype-fitness landscape and thus predict the direction of evolution for each specific cell state of resistance. Thus, combining the inferred landscape with antibiotic resistance monitoring may allow the control of drug resistance.

To demonstrate an example of controlling evolution, we performed a simulation on the inferred landscapes using a gradient ascent-based algorithm. In the simulation, the time evolution of the cell state in the PCA space $\boldsymbol{x} = (\boldsymbol{x}_{PC1}, \boldsymbol{x}_{PC2})$ under phenotype-fitness landscape $F(\boldsymbol{x})$ is given as follows:

$$\boldsymbol{x}(t+1) = \boldsymbol{x}(t) + \eta \frac{\partial F(\boldsymbol{x}(t))}{\partial \boldsymbol{x}(t)} / |\frac{\partial F(\boldsymbol{x}(t))}{\partial \boldsymbol{x}(t)}| + \xi(t), \tag{3}$$

$$\langle \xi(t) \cdot \xi(\tau) \rangle = M\delta(t - \tau), \tag{4}$$

where $\eta$ denotes the step size for each time step. $\xi(t)$ is given by white Gaussian noise and $M$ is the amplitude of the noise. Parameters $(\eta, M)$ were estimated from the experimental trajectories. $\eta = 0.4$ and $M = 0.3$ were estimated from the median change in $IC_{50}$ per day observed in the 44 evolution trajectories under drug selection and in parent strains without drug selection, respectively. In S3A Fig, we show the simulated trajectories compared to the corresponding experimental trajectories to validate the estimated simulation parameters. With some exceptions (e.g., KME5 under TET selection), the simulation results show a qualitative agreement with the experimental trajectories.

To validate the simulations quantitatively, we measured the mean distance $L$ between the simulated and experimental trajectories using

$$L = \frac{1}{4}\frac{1}{N}\sum_{j}^{4}\sum_{i}^{N} l_{i,j}, \tag{5}$$

$$l_{i,j} = |\boldsymbol{x}_{\text{exp},i,j} - \boldsymbol{x}_{\text{sim},i,j,nn}|, \tag{6}$$

where $\boldsymbol{x}_{\text{exp},i,j}$ and $\boldsymbol{x}_{\text{sim},i,j,nn}$ are the coordinates of the experimental trajectory of the $j$th copy for day $i$ and the nearest neighbor to $\boldsymbol{x}_{\text{exp},i,j}$ within the simulated trajectory, respectively. We compared $L$ with Brownian motion with the same noise parameter $M$ for 20 independent runs in each environment (e.g., parent under TET selection). The estimates given in S3 Fig show that the simulations were significantly better than random trajectories (S3B Fig; $p = 4.66 \times 10^{-17}$, one-sided Mann–Whitney U test), validating the precision of our simulation parameters.

With the validated parameters, we evaluated whether we could construct the desired evolutionary trajectories using the simulation by dynamically changing the selection drugs. Fig 5 shows an example of controlling the evolutionary trajectory in the phenotypic PCA space. Here, we started from the parent strain under KM selection (see the leftmost panel). The gray boxes indicate the thresholds for the switching environments. Under the assumption of monitoring $IC_{50}$ values for the eight antibiotics used to construct the PCA space, we would be able to keep track of evolution in the space, which would allow us to know when the strain crosses over a threshold. As shown in Fig 5, by switching environments based on thresholds, we could create a cycle of evolution in the PCA space. Note that the inferred phenotype-fitness landscapes were constructed based on laboratory experiments with a limited number of genetic backgrounds, and, thus, evolutionary trajectories starting from different genetic backgrounds might deviate from these landscapes since the different combination of mutations could alter the landscape structure [46,47]. Such alterations in the landscape structure could be detected through discrepancies between the original inferred landscape and experimental trajectories, leading to novel insights of evolutionary constraints, their relations with the strain's genetic (or nongenetic) background, and the construction of a more robust phenotype-fitness landscape. We should emphasize here that inferred landscapes combined with evolutionary algorithms such as gradient ascent can generate protocols for laboratory evolution and produce various hypotheses for investigating evolutionary constraints. We believe that our inferred phenotype-fitness landscapes based on phenotypic PCA space can open new avenues for predicting and controlling evolution.

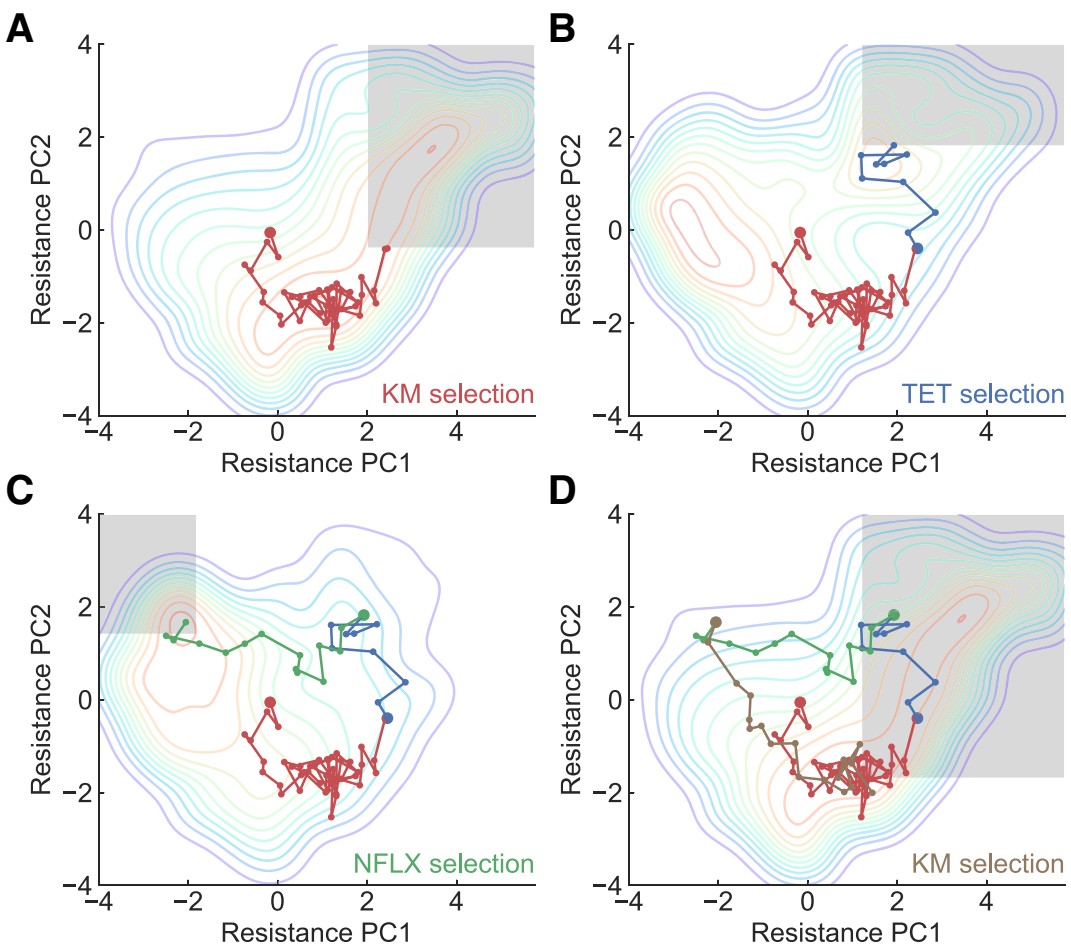

**Fig 5. Simulating an evolutionary trajectory based on the reconstructed phenotype-fitness landscape.** The gray boxes represent the thresholds set to switch environments. The title for each panel shows the name of drug used for selection. Starting from the parent strain, the sequential selection by (**A**) KM → (**B**) TET → (**C**) NFLX → (**D**) KM resulted in a circular trajectory in the PCA-projected resistance space. The data underlying this figure can be found in S1 Data. KM, kanamycin; NFLX, norfloxacin; PCA, principal component analysis; TET, tetracycline.

## Discussion

In this study, we inferred the phenotype-fitness landscape based on high-throughput measurements of the resistance profiles of eight different drugs. We demonstrated that the multiple peaks in the landscape corresponded to different antibiotic resistance mechanisms (Fig 4). We also showed that the directions of evolution predicted by the inferred phenotype-fitness landscape correspond to the directions observed in the experimental trajectories (S3 Fig). This result suggests that the inferred landscapes can predict the direction of evolution of *E. coli* at least for evolution under TET, KM, and NFLX. We also demonstrated how the inferred landscapes, combined with resistance monitoring, could control the trajectories of antibiotic resistance evolution (Fig 5). Our results are promising for predicting and controlling evolution using phenotype-based empirical fitness landscapes.

An important aspect of this study is that we inferred the fitness landscape based on antibiotic resistance profiles and not genotypes. Although previous studies have focused on constructing fitness landscapes based on mutations for a few genes [4,6], because the number of

possible genotype changes (e.g., nucleotide changes) associated with antibiotic resistance evolution is immense [32], empirical fitness landscapes based on genotypes were not capable of predicting/controlling evolution. Instead, building on recent experimental studies reporting that evolution leads to better convergence of phenotypes than that of genotypes [2], we proposed a phenotype-fitness landscape using the multidimensional antibiotic resistance profiles acquired during the course of evolution. We showed that the inferred phenotype-based fitness landscapes provided directions of evolution that were consistent with the observed trajectories, suggesting that resistance profiles can capture the internal degrees of freedom of *E. coli* for predicting evolution. Note that the space spanned by the resistance profiles corresponds to a subspace of high-dimensional space spanned by gene expression profiles [14]. Indeed, mutations conferring drug resistance have a wide spectrum and if these various mutations lead to different directions of phenotypic changes (i.e., high dimensionality of the phenotype-fitness map), it would be difficult to predict them precisely using the phenotype-fitness landscape in this study. However, previous laboratory evolution studies suggest high repeatability and a rather deterministic structure, implying the low dimensionality of the phenotype-fitness map [8,14–16,18]. For such cases, the phenotype-fitness landscape should be capable of predicting drug resistance evolution.

In this study, we used eight different antibiotics to estimate the internal state of *E. coli*. Of course, we may need more antibiotics to probe evolutionary dynamics when using different antibiotics as selection pressures owing to, for example, cryptic genetic variation [20,48]. A sufficient number of antibiotics to probe the internal degree of freedom of *E. coli* remains an open question, and we should be able to answer this by, for example, elucidating the resistance-gene expression map through extensive laboratory evolution experiments using various antibiotics with strains that have different genetic backgrounds. A related limitation of this study is that the properties of the inferred landscapes could be susceptible to the inclusion of new data, such as evolution time-series under other drugs. For example, the TET landscape has a double peak structure (Fig 4A), which is supported mainly by the eight trajectories of the "KME1 in TET" and "KME5 in TET" datasets. In S4 Fig, we show how the landscape structure changes when we remove the dataset under TET selection from the landscape inference. While the double peak structure is conserved in S4A and S4B Fig, in which a part of datasets was removed, the minor peak in S4C Fig disappeared when data in all trajectories under TET selection were removed. These results suggest that the properties of the phenotype-fitness landscape can change with the inclusion of new datasets, such as trajectories starting from evolved strains under other drugs. Because this susceptibility is caused by the lack of sufficient data, a more robust landscape is expected to be obtained through the collection of a sufficient number of experimental conditions and replicates, which should be addressed in future studies.

The inferred phenotype-fitness landscapes provided a rich space for hypothesizing the direction of evolution. In Fig 5, for example, we showed that switching antibiotic environments can lead to a cycle of antibiotic resistance states in the PCA space. Whether this evolutionary cycle is realizable in laboratory evolution systems thus needs to be determined. Although evolutionary constraints lead to repeatable outcomes, their robustness and the conditions under which constraints change remain unclear [47,49]. Changes in the genetic background during evolution may lead to different peaks in other landscapes, which could disturb the proposed evolutionary cycle. Interestingly, the inferred landscape of TET showed a small local peak that could not be observed when PCA was applied to the strains that directly evolved from the parent MDS42 strain, suggesting that the local peak in the TET landscape might have been accessible only after the acquisition of KM resistance through the decrease in PMF. Resequencing analysis, which revealed the involvement of *lon* in the local TET peak, supported this hypothesis. Taken together, our results show that changes in evolutionary constraints during

the course of evolution can be detected through inferred phenotype-fitness landscapes. Using inferred landscapes as hypothesis generators and laboratory evolution as a test ground could open a novel avenue for evolutionary biology, enabling the efficient investigation of the robustness and variation of evolutionary constraints.

## Materials and methods

### Bacterial strains and growth media

The insertion sequence (IS)-free *E. coli* strain MDS42 [50] was purchased from Scarab Genomics (Madison, Wisconsin, USA) and used as the parent strain for this study. The utilization of IS element-free strains allows reliable resequencing analyses when using short-read sequencers. In addition, to analyze evolutionary trajectories from different initial states, we used six evolved strains obtained from our previous laboratory evolution starting from the same MDS42 strain [14]. These evolved strains were isolated from the end-point culture after 27 days of serial dilutions of TET (TETE4, TETE6), KM (KME1, KME5), and NFLX (NFLXE4, NFLXE6) [14]. *E. coli* cells were cultured in a modified M9 minimal medium containing 17.1 g/L $Na_2HPO_4 \cdot 12H_2O$, 3.0 g/L $KH_2PO_4$, 5.0 g/L NaCl, 2.0 g/L $NH_4Cl$, 5.0 g/L glucose, 14.7 mg/L $CaCl_2 \cdot 2H_2O$, 123.0 mg/L $MgSO_4 \cdot 7H_2O$, 2.8 mg/L $FeSO_4 \cdot 7H_2O$, and 10.0 mg/L thiamine hydrochloride (pH 7.0) [51]. In addition, 15 μg/mL erythromycin (approximately 1/10-fold of the $IC_{50}$ of *E. coli* MDS42) was added to M9 medium throughout the experiments to avoid contamination by other bacterial species.

### Laboratory evolution

Cell cultivation, optical density (OD) measurements, and serial dilutions were performed for each antibiotic using an automated culture system [34] consisting of a Biomek NX span-8 laboratory automation workstation (Beckman Coulter, Brea, California, USA) in a clean booth connected to a microplate reader (FilterMax F5; Molecular Devices, San Jose, California, USA), shaking incubator (STX44; Liconic, Mauren, Liechtenstein), and microplate hotel (LPX220, Liconic).

Three different antibiotics, TET, KM, and NFLX, were used for laboratory evolution (Fig 1A). The antibiotics were diluted in modified M9 medium with a $2^{0.25}$-fold gradient in 384-well microplates, with 45 μL medium in each well. The $OD_{620}$ values of the precultures were measured using an automated culture system and diluted so that $OD_{620} = 0.00015$. Then, 5 μL of diluted cultures were inoculated into 45 μL of medium in each well in the 384-well microplates and cultivated under agitation at 300 rotations/min at 34°C. Every 24 h of cultivation, cell growth was monitored by measuring the $OD_{620}$ of each well, where we set $OD_{620} > 0.09$ as the parameter for cell growth. The automated culture system selected the well with the highest concentration of the antibiotic in which cells could grow for transfer. The cells in the selected well were diluted to $OD_{620} = 0.00015$ and transferred to a fresh plate containing fresh medium and antibiotic gradients.

### Quantification of $IC_{50}$

In parallel with laboratory evolution, $IC_{50}$ was determined for eight drugs: TET, KM, NFLX, SS, PLM, NQO, SDC, and MMC (Fig 1B). These eight drugs were selected to maximize the expressiveness of the phenotype space spanned by the antibiotic resistance profiles. To determine this combination of drugs, we performed a regression assessment to predict the 4,492 gene expression profiles of the 192 evolved strains in [14] from the $IC_{50}$ values of TET, KM, NFLX, and five random drugs of the corresponding strains. The combination of the five drugs

was determined through a genetic algorithm [18] using prediction accuracy as the fitness, which led to the selection of SS, PLM, NQO, SDC, and MMC. This analysis suggests that the combination of $IC_{50}$ values for the above eight drugs represents the phenotypic changes that occurred during laboratory evolution under various stress conditions.

To determine $IC_{50}$ values, serial dilutions of each drug were prepared in 384-well microplates using modified M9 medium with $2^{0.25}$-fold (TET, KM, NFLX) or $2^{0.5}$-fold (SS, PLM, NQO, SDC, MMC) chemical gradients in 22 dilution steps. Culture conditions for $IC_{50}$ determination were the same as those for laboratory evolution: $OD_{620}$ values of the cultures were measured after 24 h of cultivation in 384-well microplates containing serially diluted antibiotics. To obtain the $IC_{50}$ values, the $OD_{620}$ values for the dose–response series were fitted to the following sigmoidal model:

$$f(x) = \frac{a}{1 + \exp[b(\log_2 x - \log_2 IC_{50})]} + c, \tag{7}$$

where $x$ and $f(x)$ represent the concentration of antibiotics and observed $OD_{620}$ values, respectively. $a$, $b$, $c$, and $IC_{50}$ are fitting parameters fitted using the `optimize.curve_fit` using the SciPy package [52].

### Genome sequencing analysis

We followed the protocols for genome sequence analyses using the Illumina HiSeq System, as described in [53]. A 150-bp paired-end library was generated according to the Illumina protocol and sequenced using Illumina HiSeq (Illumina, San Diego, California, USA). Potential nucleotide differences were validated using BRESEQ (Bowtie v2.3.4.1, R v3.6.3) [54].

### Construction of the continuous landscape from discontinuous data

As shown in Fig 3C–3J, the maps of individual antibiotic resistance values in the PCA space obtained using eight-dimensional evolutionary trajectories are reminiscent of the phenotype-fitness landscape. However, this simple scatter map is still different from what can be identified as a fitness landscape for several reasons.

1. The points in the scatter map are overlapped with each other, making fitness mapping a multiple-valued function. Thus, it is difficult to use the map itself as a fitness landscape since it cannot always return a unique fitness value.

2. The points in the scatter map do not provide fitness values for unobserved points in the PCA space. Thus, we need a method that can interpolate between the observed points.

3. An important aspect of the fitness landscape is that it can provide a direction of evolution for each point in the landscape. However, our current map is just an assembly of points, and it is not easy to extract the directions of evolution from this simple description.

To solve these problems, we averaged and discretized the data and performed smoothing via a Gaussian kernel thereafter. Below, we show the details of this process and how multiple experimental trajectories of evolution were converted to an empirical phenotype-fitness landscape.

### Averaging the data

First, the resistance values were averaged from the scatter map. The entire space was discretized into square grids with a defined grid size, and the antibiotic resistance values of all points within the same grid were averaged. Here, grids with no points inside were assigned a NaN

value. Because the grid size defines the level of coarse graining in this procedure and affects the later smoothing process, we defined the grid size through the Freedman–Diaconis rule to reduce arbitrariness. The Freedman–Diaconis rule is a statistical heuristic method to select the bin width $b_w$ for constructing a histogram by

$$b_w = 2\frac{\text{ICR}\,(\text{x})}{\sqrt[3]{n}},\tag{8}$$

where ICR($x$) and $n$ are the interquartile range and the sample size of the given dataset, respectively. We applied the Freedman–Diaconis rule to the PC1 values of the data points in the PCA space, which led to a grid size of 0.519.

### Smoothing via a Gaussian kernel

Although the discretized and averaged fitness maps provide unique fitness values for the observed points in PCA space, we still could not acquire fitness values for unobserved points using this discretized map. Therefore, we applied a convolution operation to the discretized map using a Gaussian kernel (Gaussian smoothing). Here, we aimed to acquire a smoothed function $F(x)$ from the discretized and averaged map of resistance $\{f(X_j)\}_{j=1}^{N}$ where $X_j$ denotes the $j$-th grid. A smoothed function $F(x)$ can be acquired via Gaussian smoothing using

$$F(x) = \frac{1}{Z}\sum_{j=1}^{N} f(X_j)K(x - X_j, h),\tag{9}$$

$$K(z, h) = \frac{1}{h\sqrt{2\pi}}e^{-\frac{z^2}{2h^2}},\tag{10}$$

where $Z = Nh\sum_{j=1}^{N} f(X_j)$ and $h$ are the normalization factor and bandwidth of the Gaussian kernel, respectively. Here, we assumed that PC1 and PC2 are independent and thus acquired the smoothed function $F(\boldsymbol{x})$ spanning across the two-dimensional PCA space by $F(\boldsymbol{x}) = F(x_{\text{PC1}})\cdot F(x_{\text{PC2}})$. This smoothed function $F(\boldsymbol{x})$ provides fitness values for both observed and unobserved points. In addition, the derivative of $F(\boldsymbol{x})$ provided the gradient at each point in the PCA space, which could be used to predict the direction of evolution. Therefore, $F(\boldsymbol{x})$ can be interpreted as a phenotype-fitness landscape inferred from the observed evolution trajectories.

An important hyperparameter for Gaussian smoothing is the bandwidth $h$. If $h$ is too large, the fine structure of the underlying landscape may be destroyed. However, if $h$ is too small, experimental noise would dominate the inference process, worsening the predictions from the inferred landscape. Therefore, we defined $h$ after a 4-fold cross-validation. For each fold, 3/4 of the non-NaN grids in the discretized map were randomly chosen to infer the phenotype-fitness landscape using Gaussian smoothing, and their accuracy was measured by calculating the mean squared error (MSE) between the normalized resistance values of the remaining 1/4 grids. Using 4-fold cross-validation, a grid search was performed for $h$ from 0.26 to 1.5 (using the PCA space) for all eight antibiotics. As a result, $h = 0.42$ yielded the minimal MSE for the TET landscape and $h = 0.47$ yielded the minimal MSE for the other seven antibiotics. Therefore, we use $h = 0.47$ for the analysis in this study.

## Supporting information

**S1 Fig. Time series of IC$_{50}$ values in the laboratory evolution.** Data of four independent culture series are overlaid. The data underlying this figure can be found in S1 Data.
(PDF)

**S2 Fig. The inferred phenotype-fitness landscape and its validation based on the train/test splitting of the dataset.** (**A**-**C**) The inferred phenotype-fitness landscapes using the training dataset. Here, the training dataset was obtained by randomly excluding one out of the four replicates for each experimental condition, resulting in a total of 33 lines. (**D**) Differences in the evolutionary directions between the simulated gradients from the phenotype-fitness landscape ($\theta_{landscape,train}$, inferred from the training dataset) and experimental observations ($\theta_{v,test}$, based on the test dataset). Here, the test dataset contains the 11 hold-out samples that were not used in the training data. For comparison, the difference with a random direction ($\theta_{random}$) sampled from a uniform distribution $[0,2\pi]$ is also plotted. The data underlying this figure can be found in S1 Data.
(PDF)

**S3 Fig. Evolutionary simulations on the reconstructed fitness landscape.** (**A**) The simulated trajectories of evolution (gray) overlayed on the experimental trajectories (blue, orange, green, and red) in the corresponding environments. The starting points were selected randomly from the four starting points of the experimental trajectories. The simulations were performed for 10 independent runs with 10 time steps, except for the simulations for KME1 in NFLX where we ran the simulation for 150 time steps in order to let the trajectories escape from a local optimum. (**B**) The mean distances $L$ between the simulations and experimental trajectories (blue) and between random Brownian motion and experimental trajectories (orange). A total of 20 independent runs were performed for each environment, resulting into $20 \times 11 = 220$ estimates of $L$ for the simulations and Brownian motion. The data underlying this figure can be found in S1 Data.
(PDF)

**S4 Fig. The inferred phenotype-based fitness landscapes when the datasets under tetracycline (TET) selection are removed.** (**A**) Dataset of "Parent in TET," (**B**) datasets of "Parent in TET" and "KME1 in TET" (8 trajectories), and (**C**) datasets of "Parent in TET," "KME1 in TET," and "KME5 in TET" (12 trajectories) were removed from the landmark inference, respectively. KME1 and KME5 strains evolved from independent culture series under the selection of the antibiotic, kanamycin (KM). The data underlying this figure can be found in S1 Data.
(PDF)

**S1 Data. The individual numeric values in Figs 1C, 2A, 2B, 3A–3J, 4A–4I, 5A–5D, S1A–S1L, S2A–S2D, S3A, S3B and S4A–S4C.**
(XLSX)

## Acknowledgments

We thank Dr. Nen Saito and Dr. Saburo Tsuru for fruitful discussions.

## Author Contributions

**Conceptualization:** Junichiro Iwasawa, Chikara Furusawa.

**Formal analysis:** Junichiro Iwasawa.

**Funding acquisition:** Chikara Furusawa.

**Investigation:** Junichiro Iwasawa, Tomoya Maeda, Atsushi Shibai, Hazuki Kotani, Masako Kawada.

**Methodology:** Junichiro Iwasawa.

**Supervision:** Chikara Furusawa.

**Writing – original draft:** Junichiro Iwasawa, Chikara Furusawa.

**Writing – review & editing:** Tomoya Maeda, Chikara Furusawa.

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
