## [Editor Report · Decision Letter 0]

26 Aug 2022

Dear Dr Furusawa, 

Thank you for submitting your manuscript entitled "Multi-dimensional antibiotic resistance time series reveals the phenotype-based fitness landscape for Escherichia coli" for consideration as a Research Article by PLOS Biology.

Your manuscript has now been evaluated by the PLOS Biology editorial staff, as well as by an academic editor with relevant expertise, and I am writing to let you know that we would like to send your submission out for external peer review. I should warn you that although the Academic Editor was intrigued by your approach, s/he was not wholly persuaded of the utility of the fitness maps, so we will be looking for some degree of enthusiasm from the reviewers.

Once your full submission is complete, your paper will undergo a series of checks in preparation for peer review. After your manuscript has passed the checks it will be sent out for review. To provide the metadata for your submission, please Login to Editorial Manager (https://www.editorialmanager.com/pbiology) within two working days, i.e. by Aug 30 2022 11:59PM.

Kind regards,

Roli Roberts

Roland Roberts, PhD

Senior Editor

PLOS Biology

rroberts@plos.org

---

## [Decision Letter · Decision Letter 1]

13 Oct 2022

Dear Dr Furusawa,

Thank you for your patience while your manuscript "Multi-dimensional antibiotic resistance time series reveals the phenotype-based fitness landscape for Escherichia coli" went through peer-review at PLOS Biology. Your manuscript has now been evaluated by the PLOS Biology editors, an Academic Editor with relevant expertise, and by three independent reviewers.

You'll see that the reviews are broadly rather positive, and in light of these comments, which you will find at the end of this email, we are pleased to offer you the opportunity to address the comments from the reviewers in a revision that we anticipate should not take you very long. We will then assess your revised manuscript and your response to the reviewers' comments with our Academic Editor aiming to avoid further rounds of peer-review, although might need to consult with the reviewers, depending on the nature of the revisions.

IMPORTANT: The Academic Editor made the following comments (lightly edited), which you might find helpful as guidance when revising your manuscript:

"I particularly like the thoughtful comments of rev#1 and would like to see the authors' response and possible additional analyses on his/her comments 2 and 3. I also appreciate rev#2's comment 1 to explain/discuss the possible reason for the smoothness of the resistance maps of Fig. 4: if the phenotypic noise that is mentioned and reduced by using a moving average would reflect genotypic ruggedness, this should be mentioned; however, if this noise represents measurement error and the authors find no support for a rugged map, this may be discussed in the light of the several low-dimensional rugged genotype-fitness and genotype-phenotype maps that have been published. But I found all comments helpful and constructive (perhaps except rev#2's comment 2, which seems an impossible request, and I didn't understand his/her comment 3), and the authors may benefit from them to improve their ms."

**IMPORTANT - SUBMITTING YOUR REVISION**

*Resubmission Checklist*

*Published Peer Review*

*PLOS Data Policy*

*Blot and Gel Data Policy*

Sincerely,

Roli Roberts

Roland Roberts, PhD

Senior Editor

PLOS Biology

rroberts@plos.org

REVIEWERS' COMMENTS:

Reviewer #1:

In the paper, Iwasawa et al. evolve multiple strains of E. coli in 8 different drugs, and construct  PCA-based landscapes of drug resistance. These are then used to demonstrate the possibility of predicting and controlling evolution. The subject matter is topical and the approach seems promising. But there are some major issues to be addressed. Based on these, I cannot recommend  publication in the present  form.  

Major points 

1. The term "fitness landscape" is usually reserved for genotype-fitness maps. Yet, this term has been used frequently to mean phenotype-fitness maps (or phenotype-fitness landscapes). This is potentially misleading and should be rectified throughout the paper. 

2. Am I right that the PCA was done by pooling together all the data from the 44 experiments? If so, how robust are the properties of the phenotype-fitness landscapes to the inclusion of new datasets? For example, if one were to remove the lines evolved in TET, would one have qualitatively different landscapes and different predicted dynamics for the other drugs? This is important since the correspondence between theta_v and theta_landscape does not appear too strong in Fig 4I. 

3. In a similar vein, the fact that the PCA was done for the entire set of time series and then used to predict properties of the same time-series appears somewhat circular.  I understand that the point here is that a dimensionally-reduced representation still retains considerable information about the system. But the predictive power of the approach would be convincing if the training and testing datasets were distinct. For example, if two out of the four replicates were chosen randomly from each combination to obtain the landscapes, how good would the predictions be for the remainder of the datasets? 

4. There are a few issues with the section "Utilizing landscapes for controlling evolution". Firstly, the "agreement" between the simulated and experimental trajectories in Fig S2A is not convincing (the fact that the match is better than random is not very useful). Given the high noise and sample-to-sample variation in the experimental data, I am not sure why this exercise is necessary in the first place. The results inFig 5 can stand on their own. Secondly, the authors say that "Of course, real trajectories might deviate from the simulation given here because mutations accumulate during the course of evolution, and these mutations can change the structure of the landscape....However, a deviation is something we should be pleased with since it could provide novel insights into evolutionary constraints and their relations with the strain's genetic (or non-genetic) background." Here the authors seem to be suggesting that their method can be used as a tool for generating novel insights from unexpected system behavior, which is at odds with the goal of controlling evolution based on existing knowledge. Overall, the message of this section is not clear.  

Minor points

1. "Phenotypes" used in the paper are the principal components that resolve the variation in IC_50 in an 8-dimensional space. Of course, the exact contribution to the principal components of biologically interpretable phenotypes is unknown. For clarity, the sense in which "phenotype" is used in this paper (i.e. principal components) should be stated early on in the paper, perhaps even in the abstract. 

2. In the abstract, the authors say "we inferred that the fitness landscape underlies the evolution dynamics of resistance". Perhaps they mean "we inferred the fitness landscape that underlies the evolution dynamics of resistance"?

3. In Results, "eight drugs other than antibiotics" should be "five drugs other than antibiotics". 

4. In Figs 3A and 3B, the end-states of the strains should also be marked in circles (of a different color). Same applies for other figures showing trajectories. It would also help to mark the maxima in Fig 4A in Figs 3A and 3B.  

Reviewer #2:

The manuscript by Iwasawa et al. addresses an important problem in the field of antibiotic resistance evolution. Being able to predict evolutionary trajectories leading to resistance is one of the common goals in the field but has been proven to be extremely difficult due to several intrinsic difficulties such as stochastic events, size of combinatorial space, unknown genotypes, and epistasis. Authors came up with an interesting approach where they utilize the entire evolutionary trajectories of 7 evolving cultures (with 4 replicates for each) under antibiotic selection. In their analysis, they primarily use phenotypic changes to construct adaptive fitness landscapes. The paper is generally very well written and clear. Experiments and analyses are well designed and performed. I have the following major criticisms.

1. Authors carry out a PCA to construct landscapes shown in Figure 4. These are interesting landscapes but they look artificially smooth and uniform. In our experience, landscapes are almost always rugged dampening our prediction capabilities. I wonder whether the smooth landscapes are resulting due to the moving average used as described in page 6: "As the raw evolutionary trajectories were subject to experimental noise, we applied a moving average of six time points to the trajectories using a triangular window." Will the landscapes look different or less predictable if no averaging was done? Also, why the window size of 6 was chosen? 

2. The predicted landscapes shown in Figs 4 and 5 have the axis of PC1 and PC2. I understand why it is done but I did not understand what PC1 and PC2 physically/biologically mean? Do these PCs correspond to anything meaningful/helpful?

3. Evolution of resistance can be predicted and even manipulated in some very special cases. But in most cases, even the very first evolutionary step (which is almost random) such a resistance conferring mutation can steer evolution towards very unpredictable directions. I like the analysis and findings of this manuscript but there is need for further description of the limitations of authors' approach. Particularly, there are several assumptions made for creating fitness landscapes. These should be articulated in the discussion.

Reviewer #3:

This is a review of "Multi-dimensional antibiotic resistance time series reveals the phenotype-based fitness landscape for Escherichia coli." This paper presents a large set of well-designed experiments and clever analyses to understand the fitness tradeoffs that come along with drug resistant mutations. These tradeoffs are then exploited to predict how the fitness of mutants will change when exposed to different drugs. The ideas in this paper are novel and the conclusions are well supported by the data. Overall, I think this is a super cool study and it's probably the best paper I've read all year. 

My major comments are mainly to address the clarity of the study. I hope they help the authors make this nice paper even better.

1. The title is very jargony. The phrase 'multi-dimensional' is particularly vague in both the title and the abstract. I eventually realized that this refers to the multiple drugs and multiple time points. Maybe something more like: "Ultra-rich data of how fitness changes across environments and over time reveals fitness landscapes that predict evolution."

2. Also, I recommended combining the following two sentences in the abstract into a single clearer sentence: "Herein, we address this problem by inferring the fitness landscape for antibiotic resistance evolution by quantifying the phenotypic changes, that is, multi-dimensional time-series measurements of the drug resistance profile. Using the time-series data of drug resistance for multiple drugs, we inferred that the fitness landscape underlies the evolution dynamics of resistance." I recommend the following instead: "Herein, we address this problem by inferring the fitness landscape for antibiotic resistance evolution by quantifying time-series data of drug resistance for multiple drugs"

3. The introduction is usually clearly and beautifully written. However, I did have trouble with this sentence: "Parallel laboratory evolution experiments have shown that most single nucleotide and amino acid changes have varying patterns and low repeatability, implying the existence of multiple paths in genotypic space to reach a fitted phenotype." How about something like this instead, "Parallel laboratory evolution experiments have shown that different single nucleotide changes can underlie similar phenotypic changes, implying the existence of multiple paths in genotypic space to reach a fitted phenotype."

4. Figure 1A and 1B were very confusing for me. I recommend the following changes. 

a. I thought 1A was describing an evolution experiment because it is labeled "laboratory evolution with…". But now I realize panel 1B actually describes the evolution. Can you switch the order so that 1B is first (make 1B the new 1A). This would help because first you do the evolution, and then at every(?) timepoint of the evolution you do the cross-resistance test.

b. Can you rename 1A (which I think should actually be 1B) with a title that makes clear this is NOT the setup of the evolution experiment, but rather the set-up of the cross-resistance measurements?

c. Can you provide more details of the set up of the evolution experiment in the text? You mention 7 starting strains, maybe also mention how many generations there are between every timepoint, and how you transfer the cells from one timepoint to the next, and whether this was done in flasks, tubes, or plates? I know this is all in the methods, but it helps to have a little bit more in the main text. It would certainly have helped me in that I would not have confused the evolution experiment with the cross-resistance test. 

d. By the time I got to 1C I understood and wrote on the figure, "wow, this is cool!"

5. I like figure 2 and agree with the conclusion that mechanisms are modular for KM and NFLX.

6. In figure 3 I made a note about the KM evolved strains finding a different peak than the parent strain. I was happy to see that this is addressed later in the manuscript. 

7. In figure 4, two of the axes are the same axes as those in figure 3. But they have slightly different names. In figure 3 they are labeled "PC1" and in 3 they are labeled "resistance PC1". Can you use the same names to make clearer that these are the same things?

8. I was confused about which conditions were analyzed in the manner depicted by figure 4I. Since you did evolutions in KM and TET only, does this analysis only work on those conditions? In other words, the landscapes are only being climbed for TET and KM. In the other cases, since there was no evolution performed, you should not expect any climbing to happen, just moving around. If I've interpreted this correctly, perhaps make clearer in the results section that this only applies to some conditions.

9. Figure 5 and the simulations are very clever. Predicting and driving evolution like this is a big deal. I hope you plan to perform similar experiments in the wet lab. That would be quite the trick! (though totally unnecessary for this manuscript, which is already a major contribution to the field).

---

## [Editor Report · Decision Letter 2]

7 Nov 2022

Dear Dr Furusawa,

Thank you for your patience while we considered your revised manuscript "Multi-dimensional antibiotic resistance time series reveals the phenotype-based fitness landscape for Escherichia coli" for publication as a Research Article at PLOS Biology. This revised version of your manuscript has been evaluated by the PLOS Biology editors and the Academic Editor.

Based on our Academic Editor's assessment of your revision, we are likely to accept this manuscript for publication, provided you satisfactorily address the following data and other policy-related requests.

IMPORTANT: Please address the following:

a) We think that your current title will be difficult for some of our readers to understand, and we wonder if you could chose something more accessible? Some of my colleagues suggested the following, but I'm not sure whether this captures the advance correctly: "Analysis of the evolution of resistance to multiple antibiotics enables prediction of the Escherichia coli phenotype-based fitness landscape." We are open to suggestions, and it might be helpful for you to consult some of your non-specialist colleagues.

b) Please provide a blurb, according to the instructions in the submission form.

c) Please address my Data Policy requests below; specifically, we need you to supply the numerical values underlying Figs 1C, 2AB ,3ABCDEFGHIJ, 4ABCDEFGH, 5ABCD, S1ABCDEFGHIJKL, S2ABCD, S3AB, S4ABC, either as a supplementary data file or as a permanent DOI’d deposition.

d) Please cite the location of the data clearly in all relevant main and supplementary Figure legends, e.g. “The data underlying this Figure can be found in S1 Data” or “The data underlying this Figure can be found in https://doi.org/XXXX”

We expect to receive your revised manuscript within two weeks. 

*Published Peer Review History*

*Press*

Sincerely,

Roli Roberts

Roland Roberts, PhD

Senior Editor,

rroberts@plos.org,

PLOS Biology

DATA POLICY:

Regardless of the method selected, please ensure that you provide the individual numerical values that underlie the summary data displayed in the following figure panels as they are essential for readers to assess your analysis and to reproduce it: Figs 1C, 2AB ,3ABCDEFGHIJ, 4ABCDEFGH, 5ABCD, S1ABCDEFGHIJKL, S2ABCD, S3AB, S4ABC. NOTE: the numerical data provided should include all replicates AND the way in which the plotted mean and errors were derived (it should not present only the mean/average values).

DATA NOT SHOWN?

---

## [Editor Report · Decision Letter 3]

16 Nov 2022

Dear Dr Furusawa,

Thank you for the submission of your revised Research Article "Analysis of the evolution of resistance to multiple antibiotics enables prediction of the Escherichia coli phenotype-based fitness landscape" for publication in PLOS Biology. On behalf of my colleagues and the Academic Editor, Arjan de Visser, I'm pleased to say that we can in principle accept your manuscript for publication, provided you address any remaining formatting and reporting issues. These will be detailed in an email you should receive within 2-3 business days from our colleagues in the journal operations team; no action is required from you until then. Please note that we will not be able to formally accept your manuscript and schedule it for publication until you have completed any requested changes.

Sincerely, 

Roli Roberts

Senior Editor

PLOS Biology

rroberts@plos.org